# From Denoising to Refining: A Corrective Framework for Vision-Language Diffusion Model

## Abstract

Discrete diffusion models have emerged as a promising direction for vision-language tasks, offering bidirectional context modeling and theoretical parallelization. However, their practical application is severely hindered by a train-inference discrepancy, which leads to catastrophic error cascades: initial token errors during parallel decoding pollute the generation context, triggering a chain reaction of compounding errors and leading to syntactic errors and semantic hallucinations. To address this fundamental challenge, we reframe the generation process from passive denoising to active refining. We introduce **ReDiff**, a **re**fining-enhanced **diff**usion framework that teaches the model to identify and correct its own errors. Our approach features a two-stage training process: first, we instill a foundational revision capability by training the model to revise synthetic errors; second, we implement a novel online self-correction loop where the model is explicitly trained to revise its own flawed drafts by learning from an expert's corrections. This mistake-driven learning endows the model with the crucial ability to revisit and refine its already generated output, effectively breaking the error cascade. Extensive experiments demonstrate that ReDiff significantly improves the coherence and factual accuracy of generated content, enabling stable and efficient parallel generation far superior to traditional denoising methods.

## 1 Introduction

Discrete diffusion models have recently emerged as a promising alternative to the dominant autoregressive (AR) paradigm for vision-language models (VLMs) (You et al., 2025; Yang et al., 2025; Li et al., 2025a; Wang et al., 2025a; Swerdlow et al., 2025; Li et al., 2025b; Yu et al., 2025). Unlike AR models, which generate text token-by-token in a fixed unidirectional manner, diffusion models conceptualize generation as an iterative denoising process. This approach allows for bidirectional context modeling, granting them greater flexibility in controlling the generation process and a theoretical potential for massive parallelization, promising significant gains in inference efficiency (Nie et al., 2025; Ye et al., 2025; Song et al., 2025; Wu et al., 2025).

However, a significant gap exists between the theoretical promise and the practical reality of these models. Existing discrete diffusion models (Nie et al., 2025; You et al., 2025; Li et al., 2025a) are often plagued by incoherent and hallucinated artifacts (e.g., formatting errors like sequential commas or visual misaligned text) when parallel generation, frequently defaulting to one-token-per-step decoding process. We argue that these shortcomings are symptoms of a deeper, more fundamental problem: the error cascade driven by a train-inference discrepancy. Models are trained exclusively on clean, ground-truth data but are required at inference to generate from their own noisy, intermediate outputs. In a parallel decoding scenario, this discrepancy becomes catastrophic. As illustrated in Figure 1 (a), an error in a few tokens instantly pollute the context for all other tokens being generated in parallel, initiating a cycle of compounding errors, produce a detailed yet entirely fabricated description of the input image.

To break this vicious cycle, we propose a paradigm shift: from passive denoising (mask recovering under fixed context) to active refining. We introduce a corrective framework for vision-language diffusion models, called Rediff, which systematically teaches the model to identify and correct its own

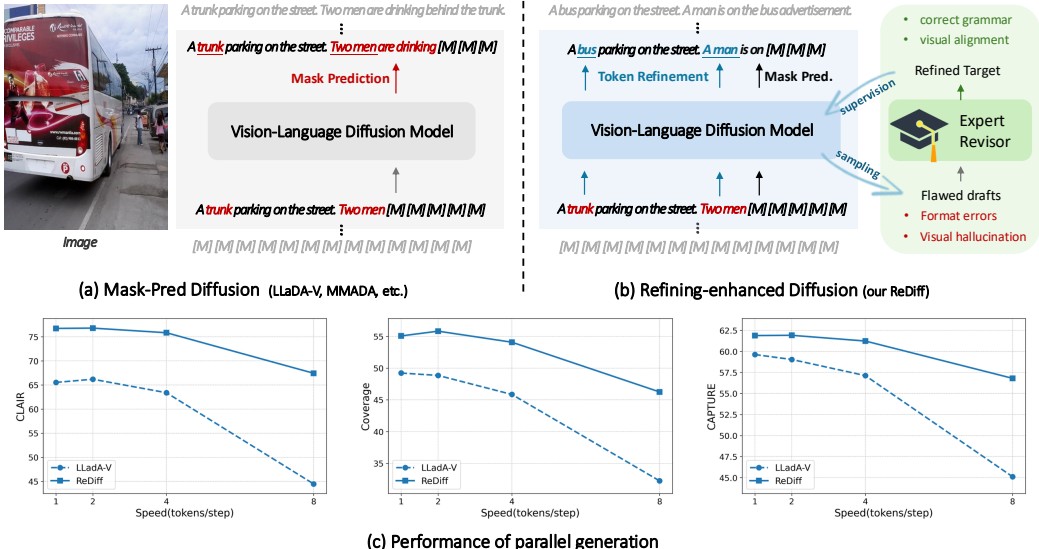

Figure 1: Comparison between standard vision-language diffusion models and our proposed refining-enhanced approach. (a) *Mask-pred diffusion* are trained for passive denoising (mask recovering under fixed context). An initial error, such as misidentifying the "bus" as a "trunk", triggers an error cascade. The model cannot correct this mistake and proceeds to hallucinate further details based on the flawed context (e.g., "Two men are drinking"), leading to a factually incorrect description. (b) *Refining-enhanced diffusion* introduces a paradigm of active refining, teaching the model not only to predict masked tokens but also to perform token refinement. ReDiff learns to self-correct through an online loop where its own "flawed drafts" are revised by an expert revisor. As a result, the model can identify and correct its initial mistake (revising "trunk" to "bus", "Two men" to "A man"), breaking the error cascade and generating a factually grounded response. (c) Performance comparison between LLaDA-V and ReDiff under different inference speeds. Our model delivers superior generation quality and achieves more stable results when using fewer inference steps.

errors during denoising. Unlike previous models that merely fill masked tokens, ReDiff actively refines the entire context to guide the generation process. Our approach consists of a two-stage training process. First, we instill a foundational revision capability by training the model to correct synthetic errors, such as random token corruptions and injected hallucinations, moving beyond simple denoising to build a general capacity for revision. Second, we introduce an online self-correction loop where the model is forced to confront and learn from its own mistakes. By capturing its flawed "drafts" during training and learning to predict an expert's revision, the model directly mitigates the train-inference discrepancy.

This mistake-driven learning endows the model with a crucial, previously absent capability: the ability to revisit and refine its own outputs, including previously unmasked tokens. By learning to self-correct, our model develops robustness to its own imperfections, effectively breaking the error cascade and enabling robust parallel generation. As shown in Figure 1 (b), our refinement-based model successfully identifies and revises an initial error, leading to a more factually grounded and accurate caption. Our contributions are threefold:

1) We propose a new perspective that reframes the generation process of diffusion models from passive denoising to active, iterative refining to address the core challenge of error cascades.

2) We design and implement a two-stage training framework, featuring a core online self-correction loop that enables the model to learn to fix its own intrinsic errors.

3) Extensive experiments demonstrate that our method significantly improves the coherence and factual accuracy of generated content, exhibiting stability far superior to traditional denoising methods, especially in challenging few-step parallel generation scenarios, thereby greatly enhancing inference efficiency.

## 2 RELATED WORK

### 2.1 LARGE LANGUAGE DIFFUSION MODELS

Discrete diffusion models (Austin et al., 2021; Lou et al., 2024; Huang et al., 2025; Arriola et al., 2025; Sun et al., 2023; Sahoo et al., 2024) represent a class of generative models tailored for discrete data like text. In contrast to image diffusion models, which corrupt data by adding Gaussian noise towards a standard Gaussian prior, text diffusion models typically operate by replacing original tokens to degrade semantic content. Early approaches, such as D3PM (Austin et al., 2021), employed discrete Markov chains where a transition matrix is progressively applied to the input, corrupting it towards a uniform distribution (i.e., any token becomes any other with equal probability) or an absorbing state (e.g., a [MASK] token). More recently, mask-and-pred diffusion models have demonstrated significant empirical success. For instance, LLaDA (Nie et al., 2025) achieves performance comparable to autoregressive large language models by generating sentences from a fully masked sequence, progressively unmasking tokens with the highest confidence. Similarly, Dream (Ye et al., 2025) has shown strong results by initializing its parameters from a pre-trained autoregressive model.

Theoretically, discrete diffusion models offer advantages over traditional autoregressive models (Touvron et al., 2023; Team, 2025; Bi et al., 2024; vicuna, 2023; OpenAI, 2023). Their bidirectional context modeling enables flexible and controllable generation, while their inherent parallelism promises significant acceleration in sampling speed. However, this potential for parallel generation remains largely untapped. Current models often suffer from output degradation—such as repetition and grammatical errors—when attempting to predict multiple tokens per step. Our work directly addresses this by enhancing the stability of parallel decoding. This aligns with a recent line of work exploring the correction of generated content. For example, SEED-Diffusion (Song et al., 2025) introduced an "Edit-based Forward Process" for code generation, which adds edit-specific noise in the final 20% of steps to allow for revisions. Likewise, FUDOKI (Wang et al., 2025a), a multimodal model based on discrete flow matching, progressively revises a random sentence, where each word is uniformly sampled from the vocabulary, to the correct answer. Our method is distinct in that it treats revision not as another form of noise, but as a high-level refinement process. Specifically, our framework trains the model to learn from and correct its own characteristic errors, moving beyond simple noise reversal.

### 2.2 LARGE VISION LANGUAGE MODELS

Large vision language models (LVLMs) (Liu et al., 2023; Dai et al., 2023; Li et al., 2024; Bai et al., 2023; Ji et al., 2023; Wang et al., 2025c) have achieved remarkable success in vision understanding and have been applied to a myriad of real-world scenarios (Ji et al., 2025; Zhang et al., 2023; Cheng et al., 2024). The dominant architecture connects a pre-trained vision encoder (Radford et al., 2021; Tschannen et al., 2025) to an autoregressive language model via a lightweight module like an MLP or Q-Former. These models first realize cross-model alignment with pre-training and then conduct visual instruction tuning to handle a wide range of vision-centric tasks.

Despite their success, a persistent challenge in LVLMs is the phenomenon of hallucination (Bai et al., 2024), where the model generates text that is factually inconsistent with the visual input. In autoregressive models, this issue is exacerbated by error propagation; an incorrectly generated token can irreversibly misguide the subsequent generation path. Notably, current multimodal discrete diffusion models, such as LLaDA-V (You et al., 2025), LaViDa (Li et al., 2025a), and MMaDA (Yang et al., 2025), also adhere to this limitation, fixing tokens in place once they are unmasked. Our ReDiff, however, leverages the bidirectional attention mechanism inherent to the diffusion paradigm. This allows our model to revisit and optimize already-generated content, enabling a progressive refinement process that directly mitigates hallucination.

## 3 METHODOLOGY

In this section, we introduce our refining-enhanced diffusion framework, ReDiff, designed to enhance the generation accuracy and stability of vision-language diffusion models. In contrast to traditional approaches that focus on recovering text from all [MASK] noise, our work emphasizes

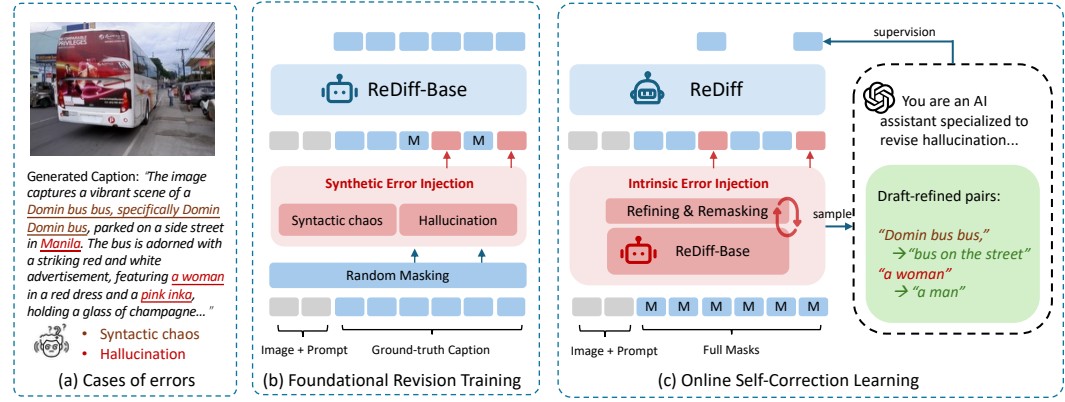

Figure 2: Overview of our proposed two-stage training framework for corrective refining. (a) We illustrate common failure modes in standard vision-language diffusion models, which are prone to generating syntactic errors (e.g., "Domin bus bus") and factual hallucinations (e.g., "a woman"). (b) In the foundational revision training stage, we instill a general corrective capability by training a base model (ReDiff-Base) to revise synthetic errors that are intentionally injected into ground-truth captions. (c) For the second stage, i.e., online self-correction learning, the model generates its own flawed "drafts". These drafts, containing the model's intrinsic errors, are then revised by an expert AI assistant. The resulting "draft-refined pairs" provide strong supervision, teaching our final model (ReDiff) to identify and correct its own characteristic mistakes, thus breaking the error cascade.

the high-level refinement of generated text. Guided by an expert model, our framework enables the model to learn from its own generation errors. This fosters a self-correction capability during inference, allowing it to simultaneously unmask new tokens while refining previously generated ones, thereby mitigating the problem of error cascades in parallel generation.

We will first present the preliminaries of discrete diffusion models in Section 3.1. We then introduce the first stage of our approach, foundational revision training, in Section 3.2 . Section 3.3 details the core of our framework, online self-correction learning. Section 3.4 details the inference process.

### 3.1 PRELIMINARIES OF DISCRETE DIFFUSION MODELS

A discrete diffusion model formalizes text generation through a forward and a reverse process. The forward process gradually corrupts a clean text sequence $x_0$ into a noisy state $x_t$ over a series of timesteps $t \in [0, 1]$. In mask-pred models, this is achieved by replacing tokens with a [MASK] token based on a noise schedule $\gamma_t$, culminating in a fully masked sequence as a prior distribution. The forward process is formulated as:

$$q\big(x_t[i] = c \,\big|\, x_0[i]\big) = \begin{cases} 1 - \gamma_t, & \text{if } c = x_0[i], \\ \gamma_t, & \text{if } c = \mathbf{M}. \end{cases} \tag{1}$$

The reverse process aims to reverse this corruption. Starting from a fully masked sequence, the model iteratively predicts the original tokens. At each step, it predicts probabilities for all masked positions, unmasks a few high-confidence tokens, re-masks the rest, and feeds the updated sequence back into the model for the next iteration.

The model, a parametric mask predictor, is trained to predict all masked tokens (denoted by a set $\mathbf{M}$) simultaneously. The training objective is a cross-entropy loss computed only on the masked tokens:

$$\mathcal{L}_{\text{CE}}(\theta) = -\mathbb{E}_{t,v,p_0,r_0,r_t} \left[ \frac{1}{t} \sum_{i=1}^{L_{r_0}} \mathbf{1}[r_t^i = \mathbf{M}] \log p_\theta(r_0^i | v, p_0, r_t) \right], \tag{2}$$

where $v$ and $p_0$ denote visual content and prompt, $r_0$ is the correct response, $t$ is sampled uniformly, and $r_t$ is sampled from the forward process.

A key advantage of discrete diffusion models is their potential for parallel generation, where multiple tokens are unmasked in a single step, significantly reducing the number of required iterations. However, existing models treat already-unmasked tokens as fixed conditions for future predictions. If an incorrect token is generated, it can derail subsequent steps, leading to an error cascade. Yet, unlike the unidirectional attention in AR models, the bidirectional attention mechanism inherent to these models provides the architectural foundation for updating previously generated tokens, a potential we exploit in our framework.

### 3.2 STAGE I: FOUNDATIONAL REVISION TRAINING

Observations of existing vision-language diffusion models, especially in few-step generation scenarios, reveal two predominant error types: syntactic chaos (e.g., incoherence, repetition, grammatical errors) and semantic hallucinations (content that contradicts the visual input), as shown in Figure 2 (a). In this first training stage, we teach the model to correct these two types of errors, extending its capability from simple denoising to foundational text revision.

We use two data construction ways. For syntactic errors, we corrupt the text from ground-truth image-text pairs by randomly replacing a fraction of tokens with other tokens from the vocabulary, creating syntactically chaotic inputs. For hallucination errors, we leverage the existing hallucination dataset ViCrit, which provides pairs of correct captions and captions with factual errors (e.g., incorrect objects, attributes, or counts). This directly provides examples of visually inconsistent text.

As shown in Figure 2 (b), we task the model with restoring a"polluted" response $r_t$ to its original, correct version $r_0$. We first apply the standard masking process to $r_0$ according to a sampled noise level $t$. Then, on the remaining unmasked tokens, we inject the synthetic errors described above. This corrupted sequence serves as the model's input. The model is trained to predict the entire original text $r_0$. The loss is computed not only on the [MASK] tokens but also on the syntactically corrupted tokens ($\mathcal{L}_{\text{syntax}}$) and hallucinated tokens ($\mathcal{L}_{\text{hallucination}}$). We also include a loss on the uncorrupted tokens ($\mathcal{L}_{\text{clean}}$) to encourage the model to preserve correct content. The loss of each type is calculated as follows:

$$\mathcal{L}_{\text{type}}(\theta) = -\mathbb{E}_{t,v,p_0,r_0,r_t} \left[ \frac{1}{t} \frac{1}{N_{\text{type}}} \sum_{i=1}^{L_{r0}} \mathbf{1}[r_t^i \in \text{type}] \log p_\theta(r_0^i | v, p_0, r_t) \right], \tag{3}$$

where $\text{type} \in \{\text{mask}, \text{syntax}, \text{hallucination}, \text{clean}\}$. Each loss component is normalized by the number of its corresponding tokens $N_{\text{type}}$ to balance their contributions. The total loss is:

$$\mathcal{L}_{\text{revision}} = \mathcal{L}_{\text{mask}} + \mathcal{L}_{\text{syntax}} + \mathcal{L}_{\text{hallucination}} + \mathcal{L}_{\text{clean}}. \tag{4}$$

After Stage I, we obtain ReDiff-Base, a model equipped with the foundational capability to correct both syntactic errors and factual hallucinations. However, this stage has a limitation: the errors are synthetic and may not reflect the characteristic mistakes the model itself is prone to making.

### 3.3 STAHE II: ONLINE SELF-CORRECTION LEARNING

To teach the model to fix its own idiosyncratic errors, we introduce an online self-correction learning framework. The process, illustrated in Figure 2 (c), proceeds as follows: (1) Generating drafts: We use ReDiff-Base to generate a response for an image, denoted as $r_{\text{draft}}$. We typically use decoding results of different generation steps to cover more mistakes. (2) Expert revision: The image $I$, the generated draft $r_{\text{draft}}$, and the ground truth are fed to a powerful external "expert model" (e.g., o4-mini). With a carefully designed prompt, the expert model identifies and corrects both grammatical and hallucinatory errors in $r_{\text{draft}}$, producing a refined version, $r_{\text{refined}}$. We specifically extract the pairs of erroneous and corrected segments. (3) Learning to refine: We form a new training instance $<I, r_{\text{draft}}, r_{\text{refined}}>$ and fine-tune our model on these data. Note that the training loss is computed only on the segments that the expert model identified and corrected. This targeted learning prevents the model from being penalized for other potential errors in the draft that the expert may have missed. The training loss is:

$$\mathcal{L}_{\text{refine}}(\theta) = -\mathbb{E}_{t,v,p_0,r_{\text{draft}},r_{\text{refined}}} \left[ \frac{1}{N_{\text{mistake}}} \sum_{i=1}^{L_{r0}} \mathbf{1}[r_{\text{draft}}^i \in \text{mistake}] \log p_\theta(r_{\text{refined}}^i | v, p_0, r_{\text{draft}}) \right]. \tag{5}$$

Table 1: Performance comparison with state-of-the-art models on three detailed image caption benchmarks. The best scores of vision-language diffusion models are in **bold**.

| Model | CapMas | | | CapArena | DetailCaps-4870 |
| | CLAIR | Coverage | Factuality | CapArena-Auto | CAPTURE |
|---|---|---|---|---|---|
| ***AR model*** | | | | | |
| LLaVA-1.5-7B (Liu et al., 2024) | 62.10 | 34.30 | 52.80 | -94.00 | 51.08 |
| InternVL-2.5-7B (Chen et al., 2024) | 78.37 | 52.57 | 78.69 | -29.83 | 57.80 |
| Qwen2.5-VL-7B Bai et al. (2023) | 80.48 | 57.32 | 82.73 | -16.83 | 60.61 |
| ***Discrete diffusion model*** | | | | | |
| MMaDA (Yang et al., 2025) | 35.45 | 14.33 | 57.98 | -97.00 | 19.55 |
| FUDOKI (Wang et al., 2025a) | 51.94 | 39.18 | 46.04 | -98.83 | 57.92 |
| LaViDa (Li et al., 2025a) | 56.22 | 44.18 | 53.57 | -90.00 | 57.28 |
| LLaDA-V (You et al., 2025) | 65.54 | 49.22 | 61.06 | -77.17 | 59.62 |
| ReDiff (ours) | **76.74** | **55.07** | **63.29** | **-51.50** | **61.88** |

To maintain the foundational capabilities learned in the first stage, we mix in a small amount of the Stage I data during this phase. This entire cycle can be iterated: the refined model from one round can be used to generate new drafts for the next round of expert revision and fine-tuning, progressively enhancing its self-correction ability. The key advantage here is that the model learns from its own mistakes, which is a more targeted and efficient way to improve its robustness and the stability of parallel generation.

### 3.4 INFERENCE PROCESS

Our inference process differs from that of traditional discrete diffusion models by integrating refinement into each generation step. Specifically, the process starts with a fully masked sequence. At each step, the model computes the output probability distribution over the entire vocabulary for all token positions. For masked positions, if the inference speed is $n$ tokens per step, we select the top-$n$ most confident tokens and unmask them. For previously unmasked positions, we replace the existing tokens with the newly predicted ones. This allows for the simultaneous unmasking of new content and refining of existing content. As more context is generated, previously generated tokens are iteratively updated to be more coherent and factually accurate, effectively reducing the occurrence of syntactic chaos and hallucinations.

## 4 EXPERIMENTS

### 4.1 EXPERIMENT SETTINGS

**Training Setup.** Our primary focus is on enhancing the generative capabilities of vision-language diffusion models. We select detailed image captioning as the representative task to validate our framework, although the methodology is generalizable to other generative tasks. Our model is built upon the existing LLaDA-V model, leveraging its foundational mask prediction capabilities while endowing it with the ability to refine. The training data comprises caption datasets from LLaVA-1.5 (Liu et al., 2023), ShareGPT4v (Chen et al., 2023), and the ViCrit dataset (Wang et al., 2025b), with ViCrit being particularly important as it contains pairs of correct and hallucinated descriptions. For Stage I (foundational revision training), we use a total of 260k image-text pairs, with a random token replacement probability of 0.1 for creating syntactic chaos. For Stage II (online self-correction learning), we generate approximately 10k draft-refined caption pairs in each round. The drafts are generated with 128, 32, and 16 inference steps, and o4-mini serves as the expert model for revisions. Our experiments revealed that a single round of this online refinement training yielded the most significant improvements.

**Benchmarks and Evaluation Setup.** We evaluate our model on three recent benchmarks for detailed image caption: CapMAS (Lee et al., 2024) uses three metrics evaluated by GPT-4o: CLAIR for overall caption quality, Coverage for the comprehensiveness of the description, and Factuality for the accuracy of the content. CapArena (Cheng et al., 2025) employs a pairwise comparison methodology where the outputs of the test model are compared against those of three baseline models with

Table 2: Performance comparison of different inference steps on CapMas benchmark. "Mask-pred training" indicates training with the traditional mask-pred objective using identical datasets.

| Metrics | CLAIR | | | | Coverage | | | | Factuality | | | |
|---|---|---|---|---|---|---|---|---|---|---|---|---|
| Speed (token/step) | 1 | 2 | 4 | 8 | 1 | 2 | 4 | 8 | 1 | 2 | 4 | 8 |
| LLada-V | 65.54 | 66.20 | 63.40 | 44.47 | 49.22 | 48.85 | 45.85 | 32.24 | 61.06 | **61.10** | 60.69 | 64.97 |
| Mask-pred training | 74.53 | 73.57 | 69.23 | 46.38 | 54.15 | 54.11 | 47.60 | 29.69 | 59.68 | 58.43 | 59.66 | **67.79** |
| ReDiff | **76.74** | **76.81** | **75.85** | **67.44** | **55.07** | **55.82** | **54.08** | **46.25** | **63.29** | 60.95 | 60.87 | 65.14 |

Table 3: Performance comparison of different inference steps on CapArena and CAPTURE metrics.

| Metrics | CapArena-Auto | | | | CAPTURE | | | |
|---|---|---|---|---|---|---|---|---|
| Speed (token/step) | 1 | 2 | 4 | 8 | 1 | 2 | 4 | 8 |
| LLada-V | -77.17 | -84.00 | -90.50 | -99.00 | 59.62 | 59.04 | 57.12 | 45.11 |
| mask training | -56.00 | -70.50 | -90.33 | -98.33 | 59.98 | 59.61 | 56.99 | 45.12 |
| ReDiff | **-51.50** | **-56.83** | **-72.67** | **-91.67** | **61.88** | **61.91** | **61.23** | **56.80** |

GPT-4o. A final score is calculated based on these win ratio. DetailCaps-4870 (Dong et al., 2024) uses the CAPTURE metric, which scores the generated caption by comparing its scene graph to that of the ground-truth description. We compare ReDiff against several vision-language diffusion models, including LLaDA-V, LaViDa, MMaDA, and FUDOKI. We also include results from some typical AR-based VLMs. At inference, the maximum generation length is 128. An inference process of 128 steps corresponds to a speed of 1 token/step, while 32 steps correspond to 4 tokens/step.

## 4.2 MAIN RESULTS

As shown in Table 1, our ReDiff achieves state-of-the-art results among all diffusion-based models across each metric. On CapMas, our model's CLAIR score shows a remarkable 11.2 point improvement over the LLaDA-V, reaching a comparable level to InternVL-2.5. The Coverage and Factuality scores also increase by 5.85 and 2.23 points, respectively, indicating that our captions are not only richer in content but also more accurate. On CapArena, our model outperforms LLaDA-V by 25.67 points. Furthermore, we achieve a CAPTURE score of 61.88, surpassing the powerful Qwen2.5-VL. These results demonstrate that our refining-enhanced diffusion method effectively improves fluency and mitigates hallucinations, leading to a substantial enhancement in overall caption quality.

In Tables 2 and 3, we compare models trained with the traditional mask-pred objective versus our refinement framework, using identical datasets and base model. Our model consistently outperforms the mask-trained baseline at every step count. Crucially, as the generation speed increases (i.e., fewer steps), our model's performance degrades much more gracefully, demonstrating superior stability in parallel generation. For instance, on the CLAIR metric, as the speed increases from 1 token/step to 8 tokens/step, the mask-trained model's score plummets from 74.53 to 46.38, whereas our model's score only decreases from 76.74 to 67.44. Notably, our model's performance at 4 tokens/step is higher than that of both LLaDA-v and the mask-trained baseline at 1 token/step. A similar trend is observed for Coverage. The trend for Factuality is less pronounced, as the baseline's score does not drop significantly at fewer steps. This is because the metric relies on extracting valid items for verification; as the baseline's output becomes more chaotic, fewer items can be extracted, artificially stabilizing the correctness ratio. On both CapArena and CAPTURE, our model also demonstrates more robust parallel generation, with the CAPTURE score dropping by only 0.65 points when accelerating from 1 to 4 tokens/step.

## 4.3 ABLATION STUDIES

**Impact of Each Training Stage.** In Table 4, we analyze the individual contributions of our two training stages. Both Stage I (foundational revision) and Stage II (self-correction) independently improve the model's performance and stability over the LLaDA-V baseline. Furthermore, the most significant gains are achieved when both stages are combined. Notably, Stage II alone provides a more substantial boost than Stage I, confirming that teaching the model to learn from its own intrinsic errors is a highly effective refinement strategy. After Stage I training, the model exhibits stable parallel generation performance. For example, as the speed increases from 1 to 4 tokens/step,

Table 4: Effect of each training stage in the refining-enhanced diffusion paradigm.

| Metrics | CLAIR | | Coverage | | Factuality | | CapArena-Auto | |
|---|---|---|---|---|---|---|---|---|
| Speed (token/step) | 1 | 4 | 1 | 4 | 1 | 4 | 1 | 4 |
| LLada-V | 65.54 | 63.40 | 49.22 | 45.85 | 61.06 | 60.69 | -77.17 | -90.50 |
| Base + Stage I | 71.31 | 71.67 | 51.73 | 51.83 | 58.04 | 55.22 | -69.17 | -73.17 |
| Base + Stage II | 73.02 | 73.52 | 53.44 | 53.00 | 59.49 | 57.40 | -68.00 | -77.67 |
| Stage I + Stage II | **76.74** | **75.85** | **55.07** | **54.08** | **63.29** | **60.87** | **-51.50** | **-72.67** |

Table 5: Effect of different settings in the foundational revision training stage.

| Metrics | CLAIR | | Coverage | | Factuality | | CapArena-Auto | |
|---|---|---|---|---|---|---|---|---|
| Speed (token/step) | 1 | 4 | 1 | 4 | 1 | 4 | 1 | 4 |
| LLada-V | 65.54 | 63.40 | 49.22 | 45.85 | **61.06** | **60.69** | -77.17 | -90.50 |
| Revise hallucination | 69.33 | 67.01 | 51.08 | 46.61 | 59.46 | 57.06 | -74.33 | -87.67 |
| Revise syntactic errors | 69.48 | 70.30 | **52.12** | 49.96 | 56.57 | 56.15 | -69.67 | -88.83 |
| Dynamic revise ratio | 68.26 | 67.98 | 50.49 | 48.66 | 59.23 | 56.60 | -74.83 | -82.50 |
| Ours (ReDiff-Base) | **71.31** | **71.67** | 51.73 | **51.83** | 58.04 | 55.22 | **-69.17** | **-73.17** |

CLAIR improves from 71.31 to 71.67, and CapArena changes from -69.17 to -73.17. The combination of the two stages yields a synergistic effect, with Stage I providing a foundational revision ability that is further amplified by Stage II, leading to large improvements in metrics like Factuality (+5.25) and CapArena (+17.67).

**Analysis of Foundational Revision Training.** As shown in Table 5, we investigate different settings for the stage I training. We find that revising syntactic errors primarily boosts overall quality (CLAIR) and Coverage, while also enhancing stability during parallel generation. Conversely, training on hallucination revision exhibits higher Factuality. Combining both error types allows our model to achieve the best overall performance. We also compare dynamic probability for random token replacement in the fourth line, where the dynamic rate is correlated with the noise level t (using t as replacement probability, when $t < 0.1$). The results indicate that our fixed replacement rate yields better overall performance.

**Impact of Online Self-Correction Training Rounds.** In Table 6, we examine the effect of iteration of the stage II training. The results show that while the first round of self-correction provides a substantial performance boost over the ReDiff-Base model, subsequent rounds of training on newly generated data do not yield further significant improvements across most metrics.

## 4.4 QUALITATIVE ANALYSIS

We provide qualitative examples to visually demonstrate how the refinement during inference produces more accurate and fluent results, thereby improving the stability of parallel generation.

In Figure 3, we compare the parallel-generated captions from ReDiff and LLaDA-V. The baseline's output suffers from token repetition ("bus", "the"), grammatical errors, and hallucinations (e.g., misidentifying a person on a bus advertisement as "a woman"). In contrast, our model's output is fluent, coherent, and factually grounded. In the second example, our model accurately describes all key elements in the scene, whereas the baseline's output is chaotic and omits significant details.

Figure 4 visualizes the token-level changes during a 32-step generation process. It clearly shows the model simultaneously unmasking new tokens and refining previously generated ones. For instance,

Table 6: Effect of online self-correction learning rounds.

| Metrics | CLAIR | | Coverage | | Factuality | | CapArena-Auto | |
|---|---|---|---|---|---|---|---|---|
| Speed (token/step) | 1 | 4 | 1 | 4 | 1 | 4 | 1 | 4 |
| ReDiff-Base | 71.31 | 71.67 | 51.73 | 51.83 | 58.04 | 55.22 | -69.17 | -73.17 |
| Online training round 1 | **76.74** | **75.85** | 55.07 | **54.08** | **63.29** | **60.87** | **-51.50** | **-72.67** |
| Online training round 2 | 76.10 | 74.99 | **55.20** | **54.08** | 62.24 | 60.46 | -56.17 | -72.83 |

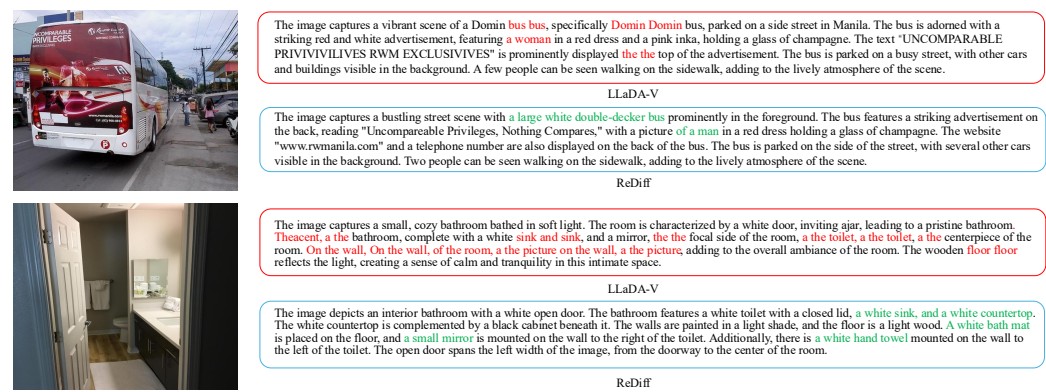

Figure 3: Cases comparison between LLaDA-V and our ReDiff under 4 tokens/step inference speed. ReDiff demonstrates superior fluency and accuracy in its generated captions.

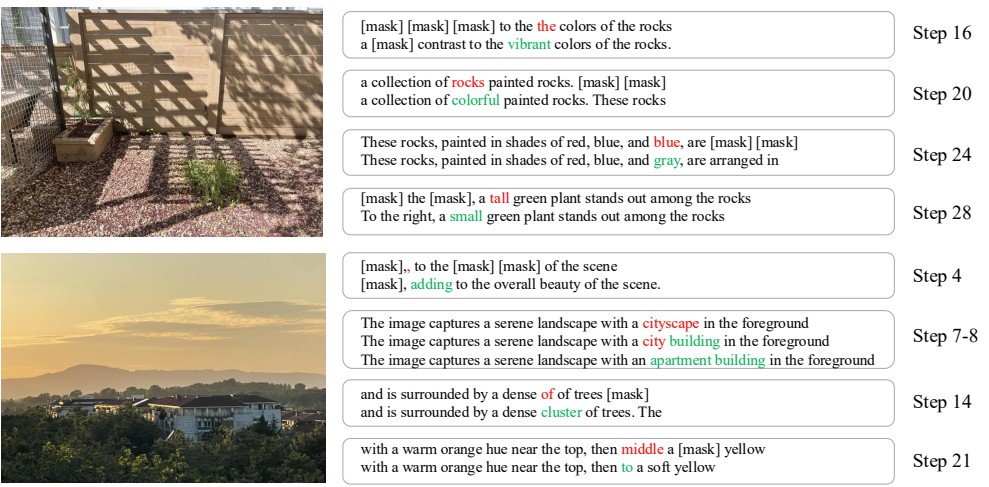

Figure 4: Refinement process of ReDiff at different inference step. Red tokens indicate the errors produced during generation, while green tokens mean the corresponding refined results.

in the first example, the model refines the erroneous phrase "rocks painted rocks" into "colorful painted rocks" at step 20. At step 28, it corrects "a tall green plant" to "a small green plant" to better match the visual content. Beyond correcting the model's own errors during generation, it also demonstrates a powerful, generalizable ability to revise disturbing inputs. More visualization of refinement can be found in Appendix A.

## 5  CONCLUSION

In this work, we addressed the critical challenge of error cascades that hampers the performance of vision-language diffusion models, particularly in efficient parallel generation scenarios. We proposed a paradigm shift from passive denoising to active refining by introducing ReDiff, a novel framework centered on a mistake-driven, online self-correction loop. This approach teaches the model to learn from its own characteristic errors, endowing it with the ability to revisit and refine its generated output. Our extensive experiments validate that this method not only achieves state-of-the-art performance but, more importantly, demonstrates far superior stability and factual accuracy in challenging few-step generation regimes where traditional denoising models catastrophically fail. By effectively breaking the error cascade, our work presents a promising path toward developing more robust, efficient, and controllable generative systems.

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
