# A More Visualization

Figure 5 showcases comparison of inference with and without the refinement, showing that the refinement is critical for achieving high-quality outputs. If ReDiff inferences without the refinement, errors tend to accumulate, such as repeated words or symbols and incoherent sentences, ultimately degrading the quality of the caption. This highlights the importance of the model's refinement capability.

Beyond correcting the model's own errors during generation, ReDiff also demonstrates a powerful, generalizable ability to revise disturbing inputs. As shown in Figure 6, we provide the model with an image and a user-provided caption containing either syntactic chaos or a factual hallucination. In both cases, our model successfully corrects the initial erroneous text and proceeds to generate a coherent and accurate completion, highlighting the strong revision ability of ReDiff.

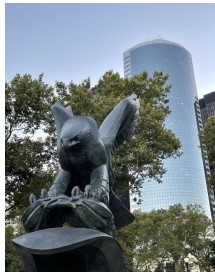

The image depicts a striking statue of a bird perched on a pedestal, situated in a city park. The bird, possibly a a eagle, is depicted in a dynamic pose, with its wings spread wide and if about to take flight... The bird's gaze is directed on the viewer, creating a sense of connection and presence.. The statue is surrounded by lush green trees,, adding a natural contrast to the scene. In the background, the building stands tall, a reflective glass glass building, typical of a the U.S. skyline. The backdrop is a clear blue sky, adding to the overall ambiance of the scene.

Inference w/o Refinement

The image depicts a striking statue of an eagle perched on a pedestal, situated in a city park. The bald eagle, rendered in gray, is poised with its wings spread wide, as if about to take flight. The statue is mounted on a square pedestal, which is a dark gray with a subtle gray section at the base. The statue is surrounded by lush green trees, adding a touch of nature to the scene. In the background, a tall, very-rise office building with reflective windows stands prominently, reminiscent of a typical U.S. skyline. The backdrop is a clear blue sky, adding to the overall ambiance of the scene.

Inference with Refinement

Figure 5: Generation results of ReDiff with or w/o refinement during inference.

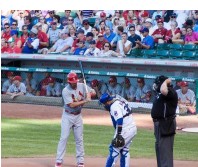

Input: a the to computer teach
Output: In the image, a baseball game is in progress with a batter, a catcher, and an umpire at home plate. The batter, dressed in a gray uniform with a red helmet, is poised to swing at the incoming pitch. The catcher, clad in a blue and white uniform with a blue helmet, is crouched behind home plate. The umpire, dressed in a black shirt and gray pants, stands just behind the catcher, observing the play. The stadium is filled with spectators, both seated and standing, dressed in various colors including red, blue, white, and black, adding to the lively atmosphere of the game.

Input: Some people are playing basketball
Output: Three people are playing baseball in a stadium. The batter, dressed in a gray uniform with a red helmet, is poised to swing at a pitch. The catcher, wearing a blue and white uniform with a blue helmet, is crouched behind the batter. The umpire, dressed in a black shirt and gray pants, is standing behind the catcher, observing the play. The stadium is filled with spectators, many of whom are wearing red and blue shirts, watching the game intently. A chain-link fence is visible along the perimeter of the field. Advertisements are visible in the stands, adding to the lively atmosphere of the game.

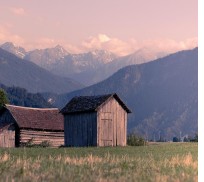

Input: house the to teach
Output: In the foreground of image, a grassy field stretches out, dotted with tall, dry grass. In the distance, two rustic wooden barns stand, their weathered wood adding a touch of rustic charm to the scene. The barns are characterized by their brown roofs, and one of them features a sturdy wooden door. In the background, a majestic range of mountains stretches across the horizon, their slopes adorned with a mix of snow and evergreen trees, creating a striking contrast against the clear sky. The mountains, with their snow-dusted peaks, glisten under a pale pink sky, adding to the serene ambiance of the scene.

Input: There are some people
Output: There are no people visible in the image. The image captures a serene rural landscape with a large, open grassy field in the foreground. In the field, there are two rustic wooden barns, one larger than the other, with a single wooden door and a dark, moss-covered roof. Small trees and bushes can be seen near the edge of the field, adding a touch of life to the scene. The background features a range of majestic mountains, some capped with snow, under a clear pink sky, enhancing the tranquil atmosphere of the scene. More trees can be seen in the distance, adding to the natural beauty of the scene.

Figure 6: ReDiff can revise wrong input answers.

# B Ethics statement

All datasets and models used in this study are publicly available and open-source. No proprietary, private, or personally identifiable information was collected or used. The images employed are either natural scenes or normal human activities, without any violent, explicit, or otherwise harmful content. Therefore, the research meets relevant considerations regarding privacy, ethics and copyright.