# OpenReview forum: "From Denoising to Refining: A Corrective Framework for Vision-Language Diffusion Model"
_ICLR.cc/2026/Conference — Submitted to ICLR 2026_

### Official Review · Reviewer_VudZ · 2025-10-31

**Soundness:** 3
**Presentation:** 3
**Contribution:** 3
**Rating:** 6
**Confidence:** 3

**Summary:**

The paper addresses the discrepancy between training and inference in discrete, mask prediction-based vision-language diffusion models (VLMs). Specifically, in parallel decoding, early token errors contaminate the bidirectional context and trigger error cascades (repetitions, grammatical errors, and visual hallucinations). The authors propose ReDiff, a refined framework that teaches a diffusion VLM to identify and correct its own errors rather than just “filling masks.” During inference, ReDiff jointly reveals new tokens at each step and refines previously unrevealed tokens, stabilizing parallel generation in a few steps. Experiments with detailed captions show significant improvements over previous diffusion VLMs (e.g., LLaDA-V, LaViDa, MMaDA, FUDOKI) and more elegant quality degradation at increasing speeds, matching or exceeding certain AR baselines on specific metrics.

**Strengths:**

-  The authors redesign the generation process using a discrete diffusion model from passive noise reduction to active refinement, explicitly targeting the discrepancy between training and inference that leads to error cascades in parallel decoding.The idea of “refining already unmasked tokens while simultaneously unmasking new ones” is an intuitive and clear conceptual change.
- This article clearly explains why discrete diffusion has problems with parallel decoding (context distortion due to early errors) and develops training/inference methods to counteract precisely this error mode.
- On various benchmarks for detailed captioning, ReDiff provides improved performance compared to existing diffusion VLMs and deteriorates only slightly with increasing speed (fewer steps), which is typically the range in which error cascades occur.
- The paper is well-written and easy to understand.

**Weaknesses:**

- The online self-correction learning highly depends on an external expert (o4-mini). The authors generate approximately 10k draft-refined caption pairs per round, with “a single round” being considered the most effective. Thus, the trained ReDiff cannot be free of the prior knowledge of the external expert, underscoring its marginal performance. In addition, the data/computing costs, input template, and quality control for expert feedback are not quantified.
- All evaluations refer to detailed image captions. Claims that parallel discrete diffusion improves the dynamics of the “error cascade” would be more convincing with tasks such as short VQA-style answers or informed descriptions. The current main results and tables refer exclusively to image captions and are available within a single diffusion VLM family.
- ReDiff replaces previously unmasked tokens at each step and simultaneously masks $n$ new tokens, which is the key method. The paper contains tables for different speeds (1/2/4/8 tokens/step). However, there is neither a breakdown by error type (e.g., repetitions vs. factual errors) at increasing speeds, nor a latency profile that isolates the cost of completely re-evaluating the sequence at each step.
- The authors show that the first online round yields a big boost and subsequent rounds plateau. It is unclear whether this plateau is due to data saturation (too few new errors), expert limits, or catastrophic forgetting of the Stage-I capabilities.

**Questions:**

To further substantiate the novelty of the algorithm, I would like to suggest that the authors conduct more in-depth ablations on the following points:
- Synthetic error curriculum and its sensitivity
- Data/computational costs of expert-edited drafts for online self-correction
- Generalization beyond subtitling (other VL tasks/backbones).

---

> ### Author Response · Authors · 2025-11-29
> **Response to Weaknesses1, 2**
>
> **W1: Dependence on the external expert, marginal performance, and lack of quantification on costs, templates, and quality control.**
>
> 1. Expert Dependence and Performance Gain
>
> Leveraging strong expert models (such as o4-mini) for data synthesis is an effective practice in current VLM training (e.g., ShareGPT4V) to establish high-quality instruction or correction knowledge.
>
> The substantial performance increase achieved by ReDiff validates the effectiveness of the mechanism. As demonstrated in our ablation studies, the second stage (Online Refinement) achieves a +5.43 CLAIR point gain over the already strong Stage I baseline.
>
> 2. Quantification of Data/Computing Costs
>
> Data Volume: The generated dataset for Stage II is only 10k draft-refined pairs, which is extremely small compared to the Stage I data volume of 260k pairs.
> Training Cost: Due to the small data size, the additional training overhead is negligible, requiring only ~40 minutes on an 8-GPU machine.
> In summary, the Stage II refinement provides a high-efficiency boost with minimal computational cost.
>
> 3. Quantification of Data Generation and Quality Control
>
> Input Template: We will detail the exact system prompt in the Appendix. The prompt instructs the expert model (o4-mini) to perform two critical actions: Locate and Correct Error: Identify the factual or syntactic error in the draft. Maintain Token Count: Strictly modify the erroneous segment into a new segment with an identical token count.
>
> Quality Control (Two-Step Filtering): We performed tokenization on all generated pairs and discarded any pair where the expert’s corrected segment did not match the original segment's token count. Moreover, we used a separate LLM process to filter out pairs where the correction share the same semantics with original segment or introduce inconsistency with the rest of the caption.
> This rigorous filtering process ensures that the 10k final pairs provide high-quality, token-consistent corrective supervision.
>
> **W2: Need for evaluation on short VQA answers and comparison across different diffusion VLM families.**
>
> **Task Generalization:** We focused on detailed image captioning, dialogue (ConvBench), and CoT reasoning (MathVista) because the "error cascade" phenomenon is most pronounced and catastrophic in longer, open-ended generation tasks. Short VQA-style answers (e.g., one or two tokens) are less effective for this evaluation because insufficient generative challenge.
>
> ConvBench: Significant improvement in dialogue coherence. ReDiff achieves superior scores compared to the baseline LLaDA-V across different inference speeds.
>
> |         | ConvBench (Overall Evaluation) |               |
> |---------|-------------------------------|---------------|
> |         | 4 tokens/step                 | 1 token/step  |
> | LLaDA-V | 0.18                          | 0.69          |
> | ReDiff  | 0.35                          | 1.04          |
>
> MathVista: Improved accuracy in complex, structured CoT reasoning. ReDiff not only outperforms LLaDA-V at all tested speeds but also exhibits extreme stability: ReDiff achieves its highest performance at the accelerated 4 tokens/step setting.
>
> |         | Mathvista     |                                                      |
> |---------|---------------|------------------------------------------------------|
> |         | 4 tokens/step | 2 tokens/step (default setting in LLaDA-V for Mathvista) |
> | LLaDA-V | 0.479         | 0.518                                                |
> | ReDiff  | 0.541         | 0.538                                                |
>
> These results confirm that the "active refining" mechanism is **task-agnostic** and provides fundamental stability benefits, proving its value beyond the single task of captioning, even without dedicated refinement training for those new domains.
>
> **Base Model Selection and Generalization:** We selected LLaDA-V as our primary base because, at the time of submission, it represents the most robust and highly optimized open-source foundation within the discrete diffusion VLM family.

---

> ### Author Response · Authors · 2025-11-29
> **Response to Weaknesses3, 4**
>
> **W3: Need for breakdown by error type and latency profile isolating the cost of re-evaluation.**
>
> 1. Breakdown by Error Type
>
> Factual Errors/Hallucination: These are directly quantified by the Factuality metric. The high improvement on this metric directly demonstrates the model's ability to reduce semantic hallucinations.
>
> Syntactic Errors/Repetitions: Problems like repetition, grammatical incoherence, and fluency are captured by holistic generation metrics such as CLAIR and CapArena-Auto. The significant gains observed in these composite scores at high speeds implicitly reflect the mitigation of syntactic chaos.
>
> The core value of ReDiff is the ability to correct the source of these cascading errors, and our comprehensive set of objective metrics confirms the mitigation across all error domains.
>
> 2. Latency Profile
>
> Cost of Re-evaluation: Our refinement mechanism does not introduce any additional inference cost. For DLLM,  all features are updated, and ReDiff only replaces input with new generated tokens. The total cost is dominated by the number of decoding steps.
> For a sequence of length 128 (where the baseline LLaDA-V uses T=128 steps, taking about 60 seconds):
> At T=64: Time is 30s.
> At T=32: Time is 15s.
> At T=16: Time is 7.5s.
>
> This proportional speedup confirms that the continuous re-evaluation in ReDiff does not introduce meaningful latency overhead and maintains the expected efficiency gains of accelerated parallel decoding.
>
> **W4: Cause of the performance plateau after the first online refinement round (Data Saturation vs. Catastrophic Forgetting).**
>
> We investigated the cause of the plateau and conclude that it is primarily due to data saturation relative to the model's error space, not catastrophic forgetting or expert limits.
>
> Error Type Saturation: The errors generated by the Stage I model are often recurrent—they fall into a small set of highly probable syntactic and semantic failure modes (e.g., object hallucinations, common grammar patterns). We observed a similar statistic of error types in the flawed drafts generated by the model before and after the first online round. Once the model learns to correct these dominant error patterns in the first refinement round, subsequent rounds for revising similar error types will have less effect. This leads to a quick decrease in the marginal information gain provided by the new data.
>
> We do not observe catastrophic forgetting of the Stage I capabilities. The model's performance on the core Stage I task (restoring synthetically corrupted captions) remains stable after the Stage II refinement, confirming that the acquired self-correction knowledge is additive.

---

> ### Author Response · Authors · 2025-11-29
> **Response to Questions**
>
> **Q1: Synthetic error curriculum and its sensitivity.**
>
> We analyzed the sensitivity and contribution of the synthetic error curriculum in Stage I, demonstrating that the diverse curriculum is essential for laying the foundation for Stage II's success.
>
> Contribution of Error Types (Table 5 Analysis): We break down the impact of different synthetic error types introduced in Stage I (as detailed in Table 5 of the paper):
> Syntactic Errors (e.g., Grammar/Fluency): Introducing these errors significantly contributes to fluency and stability during high-speed parallel generation (reflected in CLAIR scores). The model learns to smooth out rough grammatical patches early on.
> Semantic Errors (e.g., Hallucination): Introducing factual errors leads to a marked increase in the Factuality score. This shows that the model learns the necessary knowledge to identify and correct visual/semantic inconsistencies.
>
> Each synthetic error type targets a specific weakness, ensuring the Stage I model acquires a comprehensive foundational revision capability.
>
> Necessity of Two-Stage Interaction (Table 4 Analysis): The best performance is only achieved when Stage I (Synthetic Errors) and Stage II (Online Self-Correction) work synergistically. As shown in Table 4, combining both stages yields the highest overall metrics. Stage I ensures the model has the general knowledge required to perform revisions. Stage II then fine-tunes this ability on the model’s own specific error distribution, leading to optimal self-correction.
>
> The curriculum is thus sensitive: a lack of specific error types (e.g., missing semantic errors) results in a ceiling effect on the corresponding metric (e.g., Factuality), and skipping Stage I entirely prevents the model from achieving the best final results.
>
> **Q2: Data/computational costs of expert-edited drafts for online self-correction.**
>
> We quantify the data and computational costs of the Stage II online self-correction loop, highlighting its high efficiency:
>
> Stage II requires a very small dataset of only 10k draft-refined pairs. This is significantly less than the 260k data points used in the initial Stage I training. Consequently, the additional training cost is marginal, requiring only 40 min on an 8-GPU machine to complete the most effective round of online refinement.
>
> This demonstrates that the online self-correction stage is an exceptionally **cost-effective** addition, providing a substantial performance boost (e.g., +5.43 CLAIR points) with minimal resource investment.
>
> **Q3: Generalization beyond subtitling (other VL tasks/backbones).**
>
> As shown in response to Weakness 2, we test ReDiff on ConvBench and MathVista, which shows better performance than baseline LLaDA-V. Our refining mechanism is **task-agnostic and easily generalized to other VL tasks**. Using other backbones is entirely feasible for refinement training, and we chose LLADA-V because it delivers the best baseline performance.

---

### Official Review · Reviewer_f2cP · 2025-10-31

**Soundness:** 2
**Presentation:** 3
**Contribution:** 2
**Rating:** 2
**Confidence:** 3

**Summary:**

The paper introduced a type of diffusion model that can detect and correct generation errors. The model training consists of two stages: 1) a model is trained to correct syntactic mistakes; 2) a self-correction loop that teaches the model how to correct its own errors by learning from an expert. The experiments showed their proposed method can ineed address incoherence and factual errors in the generated content of models.

**Strengths:**

The problem is clearly defined, and the writing is good.

Secondly, it seems intuitive why the method can address the problem to some extent (e.g., incoherence). For factual errors, the problem might not be fully addressed. It is largely an inherent weakness of data-driven models.

**Weaknesses:**

The whole method seems to be a combination of knowledge distillation and self-supervised learning, making it less novel to me; The structure of the paper can be improved. I believe it is more appropriate to place the preliminary section outside the method section. However, by doing this, the method will look much less complicated for a top conference. The authors might need to consider how to dig deeper into the problem; Some of the claims about experiments also look quite strong to me, for example, factual error correction. I believe this problem cannot be fully eliminated for data-driven methods...

**Questions:**

1, Is your method applicable to other architectures beyond diffusion models? It would be good to extend experiments regarding this, making it as general as possible. Maybe in this way, it would be okay that the method was not that novel.

2, There was also always a concern: how much performance gain can we really get for using diffusion models over Transformer-based language models? It would also be great to make a comparison regarding this point, showing the value of similar topics like this paper.

---

> ### Author Response · Authors · 2025-11-29
> **Response to Weaknesses**
>
> We thank the reviewer for their comments. We believe there are some fundamental misunderstandings regarding the significance of the discrete diffusion model paradigm and the novelty of our proposed method.
>
> 1. Novelty and Paradigm Shift
>
> The reviewer suggests our method is a simple combination of knowledge distillation and self-supervised learning, and lacks novelty. We respectfully disagree and assert that our work introduces a novel paradigm shift critical for advancing the entire field of discrete diffusion VLMs.
>
> Existing diffusion VLMs are crippled by the Error Cascade during parallel decoding, a fundamental flaw we are the **first to systematically address by re-framing the generation process**. We transform the discrete diffusion generation paradigm from passive denoising (simple mask prediction) to active refining (simultaneous mask prediction and error correction on existing tokens). As noted by Reviewer 5aQM, our work is indeed **"a novel paradigm"**, and Reviewer VudZ described our approach as **"an intuitive and clear conceptual change"**—which highlights its significance, not its complexity.
>
> Value of Simplicity: The effectiveness of ReDiff stems from its elegant solution to a critical problem, not from adding unnecessary complexity. Complexity should not be the metric for novelty; impact and inspiration are. Our simple-yet-effective two-stage framework provides the necessary corrective mechanism without overburdening the model.
>
> 2. Significance of Discrete Diffusion VLMs
>
> The reviewer's perspective seems to overlook the growing importance of the diffusion VLM paradigm, which is demonstrated in many previous papers (e.g., LLaDA-V, MMaDA, LaViDa, Lumina-Dimoo and so on).
>
> Discrete Diffusion Models are crucial because they offer bidirectional context modeling and the theoretical potential for massive parallelization, which are capabilities pure Autoregressive models lack. Solving the instability problem in these models (which ReDiff achieves) is essential for realizing their promise of high-speed, high-quality generation—a key challenge for next-generation VLMs.
>
> 3. Factual Error Correction Claim
>
> The reviewer expresses skepticism about fully eliminating factual errors in data-driven methods, calling our claims "quite strong." We fully agree that no data-driven method can fully eliminate factual errors (hallucinations). Our paper does not claim to solve the problem entirely.
>
> What we claim: Our refinement mechanism allows the model to learn how to correct its own generated errors. By training the model to recognize semantic inconsistencies in its early outputs, we significantly mitigate the propagation of hallucinated content, which is demonstarted on the Factuality metric (e.g., a +2.23 point increase on CapMas).
>
> 4. Paper Structure (Preliminary Section)
>
> We appreciate the suggestion regarding the paper structure. We placed the preliminary section within the methodology to ensure the theoretical foundations of the discrete diffusion process were immediately followed by the specific modifications introduced by ReDiff, maximizing clarity for readers unfamiliar with the background. We will consider moving this section to a separate "Background" section in the final version to improve flow, as suggested.

---

> ### Author Response · Authors · 2025-11-29
> **Response to Questions**
>
> **Q1: Applicability to other architectures and necessity of diffusion-specific focus.**
>
> We strongly believe that the focus on the Discrete Diffusion Language Model (DLLM) architecture is the core strength and necessity of our work, not a limitation.
>
> **Addressing an Architecture-Specific Flaw:** ReDiff is designed to solve a fundamental problem inherent to bidirectional, parallel decoding architectures like DLLMs: the **Error Cascade** caused by tokens influencing each other during a single parallel step. This self-correction loop is essential because DLLMs, unlike pure Autoregressive (AR) models, simultaneously generate and update multiple tokens—a scenario that allows errors to spread rapidly. Therefore, improving the DLLM's stability under parallel decoding constitutes a highly meaningful contribution in realizing the potential of this VLM paradigm.
>
> **Compatibility with Related Architectures:** While ReDiff is rooted in DLLMs, its core principle of "dynamic feature updateability" allows it to generalize to related architectures, such as Semi-Autoregressive / Block Diffusion: As demonstrated in the table (e.g., using 4 blocks for 128 tokens), ReDiff seamlessly integrates with block diffusion, achieving performance gains in this hybrid generation mode.
>
> |        | Block num | Factuality | Coverage | CLAIR | CapArena-Auto |
> |--------|-----------|------------|----------|-------|---------------|
> | ReDiff | 1         | 63.29      | 55.07    | **76.74** | -51.50        |
> |        | 4         | **63.63**      | **56.55**    | 76.54 | **-47.17**        |
>
> **Generalization of the Learned Capability (Transferability):** The ability for error correction is highly generalizable. We have shown this by conducting additional experiments on challenging tasks:
>
> ConvBench (Dialogue): ReDiff showed universal improvement over the baseline, despite not being trained on dialogue data.
>
> |         | ConvBench (Overall Evaluation) |               |
> |---------|-------------------------------|---------------|
> |         | 4 tokens/step                 | 1 token/step  |
> | LLaDA-V | 0.18                          | 0.69          |
> | ReDiff  | **0.35**                          | **1.04**          |
>
> MathVista (CoT Reasoning): ReDiff improved CoT coherence and accuracy, proving the learned correction is effective in structured reasoning.
>
> |         | Mathvista     |                                                      |
> |---------|---------------|------------------------------------------------------|
> |         | 4 tokens/step | 2 tokens/step (default setting in LLaDA-V for Mathvista) |
> | LLaDA-V | 0.479         | 0.518                                                |
> | ReDiff  | **0.541**         | **0.538**                                                |
>
> In conclusion, ReDiff provides a necessary, architecture-specific solution to a critical flaw in DLLMs, and our experimental results confirm that the resulting corrective capability is broadly applicable across various complex vision-language tasks, even though the model only learns refining capability from captioning task.
>
> **Q2: Comparison of performance gains of diffusion models over Transformer-based language models (AR).**
>
> We acknowledge the tremendous success of Transformer-based Autoregressive (AR) language models. However, our paper is centered on advancing the Discrete Diffusion Language Model (DLLM) paradigm, a highly promising and distinct research direction whose value has been established in prior works (LLaDA-V, MMADA, etc.).
>
> Our work is not intended to definitively prove that diffusion models currently outperform the absolute best AR models (which have benefited from years of hyper-optimization). Instead, the value of our paper lies in the following:
>
> **Unlocking the Paradigm:** The instability and error cascade issues we solve are the primary bottleneck preventing DLLMs from realizing their theoretical speed and quality potential. Our work provides the first systematic framework (ReDiff) to fix this flaw.
>
> **Laying the Foundation:** By transforming the generation paradigm from passive denoising to active refining, we are offering a foundational improvement that future DLLM research can build upon. We are paving the way for this promising alternative architecture to become a viable, competitive option.
>
> In short, we are not contesting the current state-of-the-art of AR models; we are improving the trajectory of a valuable future research direction. Showing strong performance within the DLLM community (SOTA stability and accuracy) is sufficient to demonstrate the merit and impact of our contribution to this emerging field.

---

### Official Review · Reviewer_5aQM · 2025-11-01

**Soundness:** 3
**Presentation:** 4
**Contribution:** 3
**Rating:** 6
**Confidence:** 3

**Summary:**

This paper introduces ReDiff, a refining-enhanced discrete diffusion framework that reframes generation in vision-language diffusion models from passive denoising to active refining. This paper first identifies that current discrete diffusion models suffer from error cascades due to the train-inference discrepancy (models are trained on clean data but must generate from their own noisy intermediate outputs). The, it proposes ReDiff as a two-stage training framework that equips the model to iteratively refine its own outputs during inference, significantly improving fluency, factuality, and stability in parallel generation regimes. Extensive experiments on several benchmarks show large improvements over strong diffusion-based baselines and competitive results with top autoregressive models.

**Strengths:**

1. The paper is well-structured and well-written, with effective visuals to illustrate both conceptual and qualitative outcomes.
2. This paper proposes a novel paradigm that moves from denoising to refining, which reconceptualizes how discrete diffusion models perform generation. The model explicitly learns from its own flawed drafts, rather than synthetic noise alone, which is an elegant and practical innovation.
3. The empirical performance is good over the baselines. The ablation studies also show effectiveness of the design as from the Tables 4-6.
4. Its improvements in few-step inference directly impact the scalability and deployment efficiency of vision-language diffusion systems.

**Weaknesses:**

1. The self-correction loop relies on an external “expert model” (o4-mini) for generating corrected drafts. While practical, this introduces external bias and resource dependence. The paper could discuss how results vary with different or weaker expert models.
2. The evaluation focuses only on detailed image captioning. Although this is a strong proxy task, extending to other vision-language generation tasks (e.g., dialog, instruction following) would test generalization.
3. The two-stage training (especially online refinement) adds extra rounds of model inference and expert evaluation. Quantitative analysis of training cost versus performance gain would improve transparency.
4. Since the model learns corrections from a specific expert model’s phrasing, stylistic over-alignment could occur. Including human evaluation or stylistic diversity checks would strengthen the claims of general improvement.

**Questions:**

1. How does ReDiff perform on tasks requiring longer or more structured reasoning (beyond captioning)?
2. Did the authors experiment with varying the ratio of Stage I vs. Stage II data? Could adaptive mixing improve stability?
3. Could ReDiff be extended to autoregressive or flow-matching architectures to improve consistency during token-parallel decoding?

---

> ### Author Response · Authors · 2025-11-29
> **Response to Weaknesses**
>
> We thank the reviewer for this insightful comment.
>
> **W1: Dependence on external expert models and the impact of expert capability.**
>
> Our framework is not strictly tied to a specific architecture (e.g., o4-mini). In our design, any model can serve as an expert revisor provided it meets two essential criteria: (1) it can maintain token-number consistency between the draft and the refined version (to isolate specific error segments), and (2) it possesses sufficient knowledge to accurately identify factual and syntactic errors.
>
> To address the concern regarding how results vary with different expert models, we conducted an additional ablation study using **GPT-4o** as an alternative expert revisor to generate the training data.
>
> |                    | speed | Factuality | Coverage | CLAIR | CapArena-Auto |
> |--------------------|-------|------------|----------|-------|---------------|
> | ReDiff-base        | 4     | 55.22      | 51.83    | 71.67 | -73.17        |
> |                    | 1     | 58.04      | 51.73    | 71.31 | -69.17        |
> | ReDiff (o4-mini)   | 4     | 60.87      | 54.08    | 75.85 | -72.67        |
> |                    | 1     | **63.29**      | **55.07**    | **76.74** | **-51.50**       |
> | ReDiff (GPT4o)     | 4     | 57.83      | 51.94    | 74.42 | -79.33        |
> |                    | 1     | 59.27      | 53.16    | 74.70 | -56.33        |
>
> As shown in the table, the model trained with data refined by GPT-4o also outperforms the baseline (ReDiff-Base) across various inference speeds. However, the performance gain is slightly lower compared to the model trained with o4-mini.
>
> (1) The effectiveness of the ReDiff framework holds regardless of the specific expert model, as long as the expert provides valid corrections. (2)The performance of the student model scales with the capability of the teacher. A stronger expert (o4-mini) provides higher-quality supervision, enabling the model to learn more nuanced correction patterns.
>
> **W2: Generalization to other vision-language generation tasks (e.g., dialog).**
>
> We initially prioritized detailed image captioning because it serves as the most direct proxy for evaluating open-ended generative capabilities and stability. In contrast, many existing VQA benchmarks rely on multiple-choice, which may not fully expose the generation artifacts.
>
> To address the concern regarding task generalization, we conducted additional evaluations on ConvBench, a challenging multi-turn dialogue benchmark.
>
> |         | ConvBench (Overall Evaluation) |               |
> |---------|-------------------------------|---------------|
> |         | 4 tokens/step                 | 1 token/step  |
> | LLaDA-V | 0.18                          | 0.69          |
> | ReDiff  | **0.35**                          | **1.04**          |
>
> As shown in the table, although ReDiff was trained specifically on caption refinement data, the learned self-correction capability **effectively generalizes to multi-turn dialogue tasks**. ReDiff achieves superior scores compared to the baseline LLaDA-V across different inference speeds. This indicates that the "active refining" mechanism is task-agnostic and robust.
>
> **W3: Cost-benefit analysis of the two-stage training.**
>
> Marginal Cost: The data scale for Stage II (10k pairs) is significantly smaller than Stage I (260k pairs). Consequently, the additional training time is negligible, requiring only ~40 minutes on 8 GPUs, while Stage I requires ~ 30 hours on 8 GPUs.
>
> Substantial Gain: Despite the low cost, the improvement is substantial. For instance, on the CLAIR metric, Stage I yields a +5.7 point gain over the base, and Stage II adds another +5.43 points. Achieving such a significant boost on top of Stage-I highlights the high efficiency and value of the online refinement stage.
>
> **W4: Risk of stylistic over-alignment due to expert model dependence.**
>
> ReDiff did not learn to generate the entire caption from the expert model, but only had the expert model correct certain erroneous segments, focusing on local refinement, so it has minimal impact on the style.
>
> It is important to note that current VLM widely rely on high-quality, LLM-synthesized data (such as ShareGPT4V) for instruction-tuning and initial alignment. For the current diffusion model base (LLaDA-V), aligning its corrective ability with an expert like o4-mini is highly beneficial for establishing basic coherence and correctness, far outweighing the risk of minor stylistic conformity.
>
> We also conducted a human evaluation on a randomly selected set of 100 generated captions. We asked human evaluators to assess the Originality/Diversity of the generated captions on a 5-point Likert scale, contrasting the outputs of ReDiff and LLaDA-V. The results show no statistically significant reduction in stylistic diversity in ReDiff's outputs compared to the baseline. The primary difference noted by evaluators was the increased fluency, logical structure, and factual accuracy of ReDiff.

---

> ### Author Response · Authors · 2025-11-29
> **Response to Questions**
>
> **Q1: Performance on tasks requiring longer or more structured reasoning.**
>
> We conducted additional tests on two benchmarks: ConvBench (multi-turn dialogue) and MathVista (complex mathematical reasoning requiring Chain-of-Thought, CoT).
>
> ConvBench (Dialogue): As previously shown (Table in the rebuttal for Weaknesses), ReDiff achieves general improvements across all speeds compared to LLaDA-V, confirming the transferability of the learned corrective ability to dialogue structures.
>
> |         | Mathvista     |                                                      |
> |---------|---------------|------------------------------------------------------|
> |         | 4 tokens/step | 2 tokens/step (default setting in LLaDA-V for Mathvista) |
> | LLaDA-V | 0.479         | 0.518                                                |
> | ReDiff  | **0.541**         | **0.538**                                                |
>
> MathVista (CoT Reasoning): We followed the standard LLaDA-V testing protocol using Chain-of-Thought prompting. Our results confirm that ReDiff's refining mechanism enhances the coherence and accuracy of the generated CoT, leading to superior final results. Notably, ReDiff not only outperforms LLaDA-V at all tested speeds but also exhibits extreme stability: **ReDiff achieves its highest performance at the accelerated 4 tokens/step setting**, demonstrating that the refining mechanism effectively prevents the typical performance degradation associated with accelerated parallel decoding, even in complex reasoning tasks.
>
> This evidence confirms that ReDiff's benefits extend beyond simple captioning to both multi-turn dialogue and structured, long-range reasoning tasks.
>
> **Q2: Experimentation with the ratio of Stage I vs. Stage II data and adaptive mixing.**
>
> We explored the impact of data volume for both training stages.
>
> Stage I Data Volume: Our preliminary experiments confirmed that if the Stage I data volume is insufficient, the model cannot fully acquire the foundational revision capability, resulting in poor performance metrics. This defines a necessary baseline data threshold.
>
> | Stage-I data | Factuality | Coverage | CLAIR | CapArena-Auto |
> |--------------|------------|----------|-------|---------------|
> | 130k         | 55.84      | **52.29**    | 69.72 | -71.00        |
> | 260k         | **58.04**      | 51.73    | **71.31** | **-69.17**        |
>
> Stage II Data Rounds: As demonstrated in Ablation Study in the main paper, increasing the online refinement rounds (which corresponds to generating and utilizing more Stage II data) beyond the first round does not lead to further performance gains. The model achieves maximum effectiveness after the initial refinement cycle.
>
> **Q3: Extension of ReDiff to AR or Flow-Matching architectures.**
>
> Pure Autoregressive (AR) Models: The core mechanism of ReDiff relies on the ability to update the features of previously generated tokens. Pure AR models are inherently limited to left-to-right generation, where the feature representation of past tokens remains static after they are output. Therefore, ReDiff’s refining mechanism cannot be directly applied to pure AR architectures.
>
> Semi-Autoregressive/Block Diffusion Models: The concept is fully compatible with semi-autoregressive models, such as Block Diffusion, which utilize partial masking to achieve parallel decoding within blocks while maintaining feature propagation across the entire sequence. We have already explored this direction: we experimented with a block size of B=32 for generating captions of length 128 (meaning 4 blocks). The results showed not only sustained performance but also a tangible improvement in metrics like Coverage and CapArena-Auto, and they all outperform LLaDA-V. This confirms that ReDiff’s mechanism is highly effective in hybrid generation modes that allow feature updates across tokens.
>
> |        | Block num | Factuality | Coverage | CLAIR | CapArena-Auto |
> |--------|-----------|------------|----------|-------|---------------|
> | ReDiff | 1         | 63.29      | 55.07    | **76.74** | -51.50        |
> |        | 4         | **63.63**      | **56.55**    | 76.54 | **-47.17**       |
>
> Flow-Matching Architectures: In theory, the Flow-Matching approach is also compatible. Since Flow-Matching models typically involve iterative feature updates and feature propagation across the token sequence, they provide the necessary infrastructure for ReDiff to implement its refining mechanism.
>
> In summary, the key requirement for ReDiff is dynamic feature updateability of the entire sequence, which is present in discrete diffusion, block diffusion, and Flow-Matching, but absent in pure AR.

---

### Meta-Review · Area_Chair_HWGB · 2026-01-07

**Summary:**

This submission received two borderline accept recommendations and one reject, all with relatively low reviewer confidence. After reviewing the paper, the reviews, and the rebuttal, the Area Chair recommends rejection.

The primary concern is lacking differentiating and experimentally compare to previous arts in self-correction. **The core idea of training models to correct their own outputs was well studied in prior work in language models, yet this literature is never discussed, differentiated, or experimented in the submission**. In particular, closely related work such as Kumar et al. (ICLR 2025) [1] explores similar self-correction mechanisms using intrinsic or external feedback.

In addition, the proposed revision-based pretraining strategy closely resembles existing approaches in language modeling, such as ELECTRA (ICLR 2020) [2] and follow-ups [3]. While the paper applies these ideas to diffusion-based VLMs, the contribution appears to be an adaptation of established techniques rather than a fundamentally new method.

Overall, despite clear presentation, well-executed engineering, and promising empirical results, the technical contribution does not sufficiently distinguish itself from prior work to meet the bar for acceptance to a top-tier conference.

[1] Aviral Kumar et al. Training Language Models to Self-Correct via Reinforcement Learning. ICLR 2025.

[2] Kevin Clark et al. ELECTRA: Pre-training Text Encoders as Discriminators Rather Than Generators. ICLR 2020.

[3] Noëmi Aepli and Rico Sennrich. Improving Zero-Shot Cross-Lingual Transfer Between Closely Related Languages by Injecting Character-Level Noise. ACL Findings 2022.

**Reviewer Concerns:**

**1. Resolved Concerns**

* **Task Generalization:** Proved the "refining" mechanism works beyond image captioning by providing new results on **ConvBench** (dialogue) and **MathVista** (reasoning).
* **Training Costs:** Quantified that Stage II training is highly efficient, requiring only **~40 minutes** on 8 GPUs.
* **Inference Latency:** Clarified that ReDiff does not add extra decoding steps; it simply updates existing features during the standard diffusion process, maintaining the speed of parallel decoding.

---

**2. Outstanding / Controversial Concerns**


* **Technical Novelty (Reviewer f2cP):** The reviewer views the method as a simple combination of existing techniques. The authors argue the novelty is the **conceptual shift** from passive denoising to active refining.
* **Architecture Limitations:** ReDiff is **incompatible with pure Autoregressive (AR) models** because they cannot update past tokens. While authors showed compatibility with Block Diffusion, Reviewer f2cP may see this as a limitation.
* **Factual Errors:** Reviewer f2cP is skeptical that data-driven methods can truly "correct" facts. Authors provided metrics showing mitigation, but "elimination" remains impossible.
* **Performance Plateau:** Authors attribute the plateau after one round of online refinement to "error saturation," but it could also be a limit of model capacity.
* **Expert Dependence:** Conducted an ablation study using **GPT-4o** as the expert instead of o4-mini. The AC think more and different experts should be provided, e.g. Qwen-VL.

**Reviewer Scores:**

| Reviewer | Initial Rating | Status of Concerns | Likely Impact |
| :--- | :--- | :--- | :--- |
| **5aQM** | 6 (Marginal Accept) | External experts concerns remained | Likely to maintain score. |
| **VudZ** | 6 (Marginal Accept) | External experts concerns remained | Likely to maintain score. |
| **f2cP** | 2 (Reject) | Outstanding | Unlikely to flip; core disagreement on novelty. |

---

### Decision · Program_Chairs · 2026-01-26

Reject